# Interactive Effects of Nutrients and Salinity on Phytoplankton in Subtropical Plateau Lakes of Contrasting Water Depths

Ying Wang [1], Xia Jiang [2], Yan-Ling Li [1], Li-Juan Yang [1], Ye-Hao Li [1], Ying Liu [1], Long Zhou [1], Pu-Ze Wang [1], Xu Zhao [1], Hai-Jun Wang [1,*], Erik Jeppesen [1,3,4,5,6,7] and Ping Xie [1]

1  Institute for Ecological Research and Pollution Control of Plateau Lakes, School of Ecology and Environmental Science, Yunnan University, Kunming 650500, China
2  National Engineering Laboratory for Lake Pollution Control and Ecological Restoration, Chinese Research Academy of Environmental Sciences, Beijing 102218, China
3  Denmark Department of Ecoscience, Aarhus University, 8000 Aarhus, Denmark
4  Sino-Danish Centre for Education and Research, Beijing 100039, China
5  Limnology Laboratory, Department of Biological Sciences, Centre for Ecosystem Research and Implementation (EKOSAM), Middle East Technical University, Ankara 06800, Turkey
6  Institute of Marine Sciences, Middle East Technical University, Mersin 33731, Turkey
7  Donghu Experimental Station of Lake Ecosystems, State Key Laboratory of Freshwater Ecology and Biotechnology, Institute of Hydrobiology, Chinese Academy of Sciences, Wuhan 430072, China
*  Correspondence: wanghaijun@ynu.edu.cn

**Abstract:** Eutrophication and salinization are serious global environmental problems in freshwater ecosystems, occasionally acting jointly to exert harmful effects on aquatic ecosystems. To elucidate the interactive effects of nutrients and salinity on phytoplankton assemblages, we conducted a four-season study during 2020–2021 of eight lakes from Yunnan Plateau (Southwest China) with a wide range of conductivities (Cond, reflecting degree of salinization), eutrophic states, and water depths and used General Additive Modeling (GAM) of the data. We found that: (1) species number (SN), density ($D_{Phyt}$), and biomass ($B_{Phyt}$) of phytoplankton showed stronger seasonal dynamics in shallow lakes than in deep lakes, all being, as expected, higher in the warm season; (2) annual and summer data revealed highly significant positive relationships between SN, $D_{Phyt}$, and $B_{Phyt}$ with total nitrogen (TN) and total phosphorus (TP), which became weaker at high TP occurring when the N:P ratio was low, indicating N limitation; (3) SN, $D_{Phyt}$, and $B_{Phyt}$ showed a unimodal relationship with salinity, peaking at 400–1000 µS/cm (Cond); (4) the two dominant taxa (cyanobacteria and chlorophyta) showed different patterns, with chlorophyta generally dominating at low TN and cyanobacteria at high TN and Cond, suggesting the synergistic effect of nitrogen and Cond on cyanobacterial dominance.

**Keywords:** phytoplankton; eutrophication; salinization; interactive influence; water depth; plateau lakes





## 1. Introduction

Lake eutrophication is a global environmental issue [1,2], leading to proliferation of phytoplankton, deteriorated water quality, and loss of biodiversity, which eventually affects economic and cultural development [3,4]. Phytoplankton is often used as a biological indicator of the ecological and health status of a lake [5–8] due to its vital role as a primary producer and its sensitive response to various external environmental changes [9,10]. Factors affecting phytoplankton include nutrients, zooplankton, fish, light, temperature, and pH [11,12]. It is widely accepted that cyanobacteria are favored by high temperature conditions and therefore more likely to become dominant in eutrophicated lakes in a warmer climate [13,14].

As a consequence of climate warming and increasing consumption of freshwater resources, e.g., for irrigation, salinization of freshwater ecosystems has become a worldwide

challenge for the aquatic environments [15,16]. Salinization may promote phosphorus release from sediments [17], inhibit various key zooplankton grazers, and often promotes phytoplankton [18–20]. Salinization also reduces phytoplankton diversity, while cyanobacteria have a high salt tolerance to salinity and often dominate in eutrophic saline water bodies, thereby increasing the risk of cyanobacterial toxin production [21,22]. Eutrophication and salinization may act jointly to cause various adverse impacts on aquatic ecosystems [23,24].

Little is known about the role of water depth on the joint effects of eutrophication and salinization in lake ecosystems. Studies have shown that eutrophication occurs more frequently in shallow lakes and that nitrogen and phosphorus cycling is strongly determined by water depth [20,25]. In addition, phosphorus release from sediments tends to be more prominent in eutrophic shallow lakes than in deep lakes [26]. Additionally, salinity may be a factor in deep lakes that affects the stability of stratification, and this may mitigate the consequences of eutrophication [27,28]. Based on the results of previous studies, we hypothesized that nutrients and salinity may act together to promote the growth of phytoplankton [29], especially of cyanobacteria, and that phytoplankton in deep lakes might be depressed at high salinity [27,28].

Located in southwestern China, the Yunnan Plateau supports nutrient-rich lakes varying greatly in water depth. Due to excessive nutrient loading along with rapid socio-economic development, some Yunnan Plateau lakes are suffering from severe problems of eutrophication, characterized by frequent and long-duration blooms of cyanobacteria [30–32]. These lakes simultaneously face the risk of drying due to low precipitation, high evaporation, and accelerated water demand from humans, particularly from agriculture [33,34]. Consequently, the Yunnan Plateau holds a mosaic of lakes with varying degrees of eutrophication and salinization. In this study, eight large lakes were investigated to reveal the joint effects of eutrophication and salinization by comparing their phytoplankton assemblages, including species diversity, standing crops, and dominance of main taxa during four seasons.

## 2. Materials and Methods

### 2.1. Field Sampling and Measurements

The Yunnan Plateau is located in southwestern China and has an average annual temperature of 15–18 °C and an annual precipitation of 1000–1200 mm. Most of the area is situated 1500–2000 m above sea level. In this study, eight Yunnan lakes were selected for annual field investigations: Lake Dianchi, Lake Erhai, Lake Chenghai, Lake Luguhu, Lake Yilonghu, Lake Qiluhu, Lake Xingyunhu, and Lake Yangzonghai (Figure 1). These lakes range from 1503 to 2690 m in elevation, from 31 to 297 km$^2$ in area, and from 2 to 40 m in average depth. Based on their natural morphological characteristics, 7–15 sampling sites were randomly selected in each lake in each season. Quarterly sampling was conducted in October (autumn) and December (winter) 2020 and in April (spring) and July (summer) 2021.

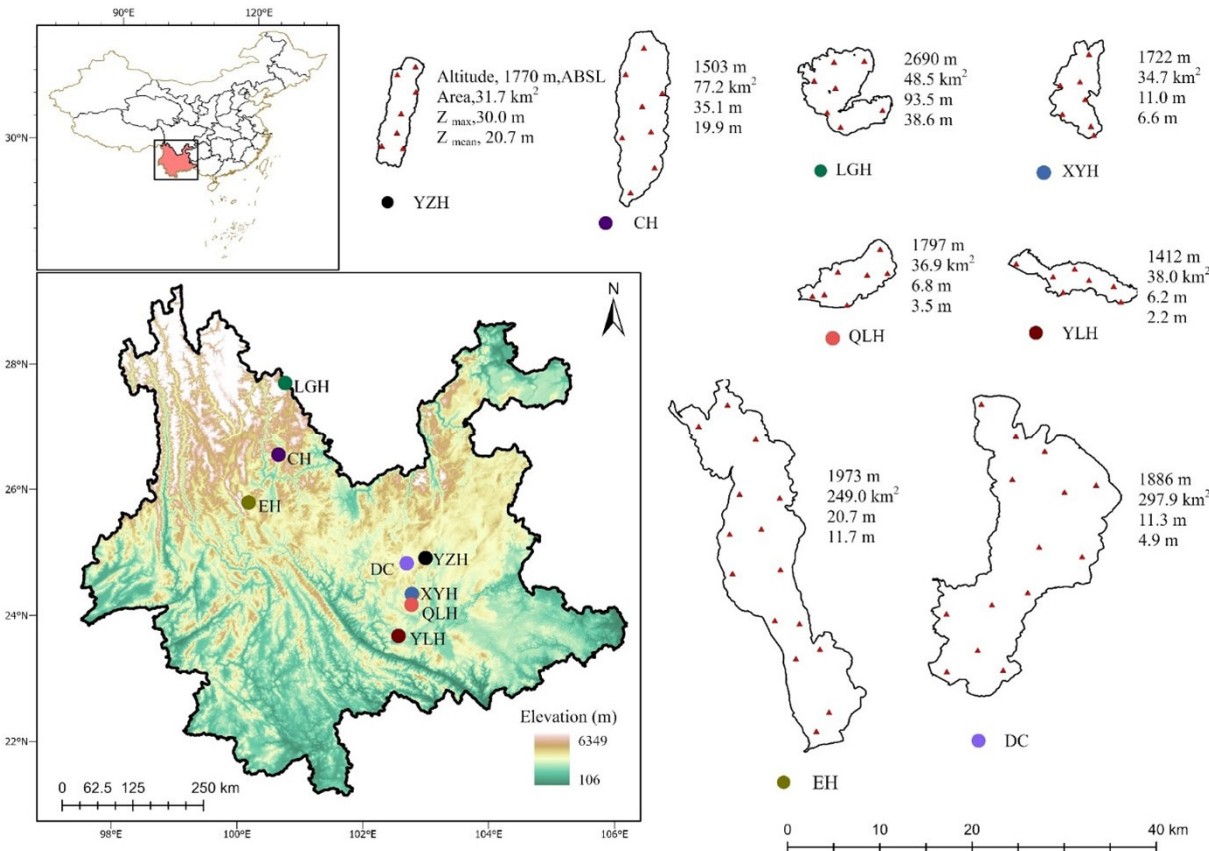

**Figure 1.** Location and sampling sites of the studied lakes. Lake Luguhu, LGH; Lake Chenghai, CH; Lake Yangzonghai, YZH; Lake Erhai, EH; Lake Xingyunhu, XYH; Lake Dianchi, DC; Lake Yilonghu, YLH; and Lake Qiluhu. QLH.

Physical variables such as water temperature (WT, °C), dissolved oxygen (DO, mg/L), pH, and conductivity (Cond, μS/cm) were measured in situ using a multiparameter water quality sonde (EXO2; Xylem, Ohio, USA). Turbidity (Turb) was measured using a HACH 2100 Q turbidimeter. Depth ($Z_M$) and transparency ($Z_{SD}$) were measured using a depth sounder and Secchi Disc, respectively. At each sampling site, water samples (5 L) were collected from containers at depths of 0.5, 3.0, and 5.0 m using column water trap samplers; 1 L mixed water samples were brought back to the laboratory for determination of other water quality variables. Phytoplankton chlorophyll a (Chl *a*), total nitrogen (TN), and total phosphorus (TP) levels were measured according to the Water and Wastewater Monitoring and Analysis Methods of the Ministry of Environmental Protection of the People's Republic of China [35]. TN was determined using alkaline potassium persulfate digestion and UV spectrophotometry (TU-1810; DPC, China); TP was determined using ammonium molybdate-UV spectrophotometry; and Chl *a* was determined spectrophotometrically after 90% acetone extraction at 4 °C for 24 h.

Furthermore, a 1 L subsample (of the 15 L pooled sample) was transferred to a polyethylene bottle and preserved with Lugol's iodine solution for phytoplankton analysis. After sedimentation for 48 h, the supernatant was aspirated with a siphon tube to a final concentrated volume of 50 mL. Analyses were conducted under a light microscope (BX 51; Olympus; 400 ×) using a 0.1 mL counting chamber [36]. Phytoplankton biomass was estimated as biovolume, and identification of the phytoplankton species was performed following the "Freshwater Algae in China-System, Classification and Ecology" [37], "Atlas of Common Freshwater Planktonic Algae in China" [38], and "Chinese Freshwater Algae Journal" [39]. The units used were cell/L for density and for mg/L for biomass.

### 2.2. Data Processing and Analyses

We used the Composite Nutritional Status Index (TLI) to assess the trophic status of the studied lakes [40] based on $Z_{SD}$ (m), Chl $a$ (µg/L), TN (mg/L), and TP (mg/L) [41]. A physicochemical index of each lake was obtained by averaging the values of each sample point in the four seasons. The trophic status of each lake was obtained by averaging the TLI values from all sampling sites in the lake. To enable comparison with other studies, the Cl concentration (mg/L) was concerted to conductivity (µS/cm) according to Moffet et al. [42] and salinity (mg/L) was converted to conductivity by dividing by 0.774 [43].

Using Origin 2022 software, Spearman rank correlation analyses were performed to determine the relationships between phytoplankton variables and the environment. Using an average water depth of 10 m as a threshold, the eight lakes were roughly divided into deep (with clear thermal stratification) and shallow lakes (without clear thermal stratification) to further analyze the potential role of water depth in influencing the response of phytoplankton to eutrophication and salinization. Due to the strong spatial heterogeneity of the water, the depth at different sampling sites in each lake, and the large distance between the sampling sites (see Figure 1), all site-specific data from the four seasons were used in the correlation analysis to reveal the potential role of the water depth. Moreover, analyses based on lake-average data for the four seasons were performed. The data were first Log$_{10}$-transformed and determined for normal distribution using a Windows-based SPSS statistical package and then analyzed with a generalized additive model (GAM) using four Windows-based R statistical packages (ggpubr, ggplot2, ggsci, and mgcv). In addition, multiple stepwise regression analysis was performed using SPSS.

## 3. Results

### 3.1. Environmental Conditions and Assemblage Structure of Phytoplankton

The trophic status and Cond (conductivity, used to reflect salinity) in the eight lakes formed a clear gradient (Table 1). The calculated TLI values suggested the occurrence of three lake groups: eutrophic (XYH, DC, YLH, and QLH), mesotrophic (CH, YZH, and EH), and oligotrophic (LGH). The measured Cond also roughly suggested three lake groups, with CH and QLH falling into the highest Cond group, YZH, EH, XYH, DC and YLH in the moderate group, while LGH constituted the lowest Cond group.

**Table 1.** Main limnological characteristics (mean ± SE) of the eight studied lakes.

| | LGH | CH | YZH | EH | XYH | DC | YLH | QLH |
|---|---|---|---|---|---|---|---|---|
| Longitude (E°) | 100.79 | 100.66 | 103.01 | 100.02 | 102.78 | 102.71 | 102.59 | 102.79 |
| Latitude (N°) | 27.72 | 26.54 | 24.91 | 25.78 | 24.34 | 24.83 | 23.67 | 24.17 |
| $Z_{SD}$ (m) | 8.21 ± 0.70 | 2.29 ± 0.37 | 1.91 ± 0.09 | 1.26 ± 0.05 | 0.46 ± 0.03 | 0.38 ± 0.03 | 0.19 ± 0.02 | 0.33 ± 0.02 |
| Turb (NTU) | 2.41 ± 1.03 | 4.29 ± 0.46 | 2.74 ± 0.39 | 5.82 ± 0.38 | 24.11 ± 1.57 | 37.77 ± 2.20 | 32.39 ± 2.16 | 24.70 ± 1.55 |
| WT (°C) | 15.91 ± 0.70 | 21.14 ± 0.69 | 19.68 ± 0.58 | 19.00 ± 0.60 | 19.77 ± 0.88 | 18.95 ± 0.60 | 21.71 ± 0.98 | 20.16 ± 0.85 |
| DO (mg/L) | 6.69 ± 0.33 | 7.31 ± 0.18 | 7.63 ± 0.20 | 7.81 ± 0.16 | 8.19 ± 0.31 | 9.99 ± 1.43 | 7.35 ± 0.33 | 9.10 ± 0.32 |
| Cond (µS/cm) | 181.8 ± 4.2 | 1236 ± 17.9 | 389.1 ± 3.8 | 277.6 ± 4.4 | 519.0 ± 11.6 | 360.9 ± 6.6 | 500.4 ± 11.2 | 880.7 ± 30.7 |
| pH | 7.83 ± 0.10 | 8.72 ± 0.05 | 8.22 ± 0.08 | 8.17 ± 0.05 | 8.31 ± 0.08 | 8.56 ± 0.06 | 8.10 ± 0.08 | 8.38 ± 0.06 |
| TN (mg/L) | 0.19 ± 0.03 | 0.77 ± 0.02 | 0.59 ± 0.02 | 0.64 ± 0.04 | 1.69 ± 0.03 | 1.69 ± 0.05 | 3.24 ± 0.12 | 3.99 ± 0.37 |
| TP (mg/L) | 0.01 ± 0.004 | 0.03 ± 0.003 | 0.02 ± 0.001 | 0.02 ± 0.003 | 0.16 ± 0.013 | 0.10 ± 0.005 | 0.06 ± 0.004 | 0.11 ± 0.004 |
| N/P | 25.8 ± 2.4 | 48.7 ± 7.6 | 31.6 ± 1.8 | 40.2 ± 4.7 | 13.2 ± 1.7 | 19.0 ± 0.7 | 59.3 ± 3.7 | 37.4 ± 2.9 |
| Chl $a$ (µg/L) | 1.89 ± 0.50 | 5.41 ± 0.50 | 13.71 ± 0.57 | 10.20 ± 0.72 | 45.45 ± 3.47 | 52.21 ± 4.24 | 72.60 ± 6.64 | 76.24 ± 9.12 |
| TLI | 15.98 ± 1.61 | 33.74 ± 1.31 | 35.12 ± 0.38 | 35.71 ± 0.61 | 51.36 ± 0.46 | 52.42 ± 0.69 | 57.13 ± 0.93 | 57.33 ± 0.57 |
| Trophic status | oligotrophic | mesotrophic | mesotrophic | mesotrophic | eutrophic | eutrophic | eutrophic | eutrophic |

Note: ZSD, Secchi depth; Turb, turbidity; WT, water temperature; DO, dissolved oxygen; Cond, conductivity; TN, total nitrogen; TP, total phosphorus; Chl a, phytoplankton chlorophyll a. Mean: the mean values of the four quarters at each sample site is the mean value of the lake. Refer to Figure 1 for explanation of lake name abbreviations.

Eight phyla were identified in the eight lakes, with chlorophyta, cyanophyta, and bacillariophyta representing the top three taxa in species number (SN). SN, density (D$_{Phyt}$), and biomass (B$_{Phyt}$) of phytoplankton showed seasonal fluctuations and were generally higher in warmer seasons (summer and autumn) than in colder seasons (spring and winter). D$_{Phyt}$ and B$_{Phyt}$ generally increased with nutrient status. The percentage of each phylum in density and biomass varied among lakes and seasons. Generally, cyanophyta ranked first in density. Mainly bacillariophyta and cyanophyta dominated biomass (Figure S1).

*3.2. Relationships of Phytoplankton with the Environments*

Spearman rank correlation analyses suggested highly significant positive relationships between phytoplankton variables (SN, $D_{Phyt}$, $B_{Phyt}$, and Chl *a*) and TN, TP, WT, and TLI, and negative relationships were suggested with $Z_M$ and $Z_{SD}$ (Figure S2). In the correlation analysis between TN, TP, Cond, and phytoplankton variables, the water depths varied significantly among sample sites, and all sampling sites in the four seasons were included in the analysis. An analysis based on the lake-average data for the four seasons was performed, showing overall similar results as the total dataset (see annexes Figures S9–S12). See Tables S1–S3 for all GAM model parameters.

The total species number of phytoplankton generally showed increasing trends with TN, TP, and Cond (Figure 2a–c). The increasing trends became sharper at high TN and gentler at high TP, while a notable decrease in SN occurred at high Cond. No clear differences were observed between shallow and deep lakes for the relationships between SN, TN, and TP. Along the gradient of Cond, a pronounced decrease in SN was observed in deep, mesotrophic lake Lake Chenghai (CH). In comparison, SN was much higher in shallow, hypertrophic Lake Qiluhu (QLH) with similar Cond (see Table 1 for basic information). SN of cyanophyta and chlorophyta showed similar dynamic trends as total phytoplankton, except for a lower SN of cyanophyta at high TN (Figure 2d–i). SN of bacillariophyta demonstrated no clear trend with TN or TP but decreased significantly with Cond, particularly in deep lakes (Figure 2j–l). The analyses for summer (Figure S5) revealed similar results as the annual data.

The two variables of phytoplankton abundance, $B_{Phyt}$ and $D_{Phyt}$, exhibited almost the same patterns of change with TN, TP, and Cond (Figures 3 and S3a–c). Generally, they increased significantly, with higher slopes at high TN, lower slopes at high TP, and markedly reduced slopes at high Cond. Similar to the changes in SN, no clear differences were found between shallow and deep lakes for changes in $B_{Phyt}$ and $D_{Phyt}$ with TN and TP. For Cond, marked decreases in $B_{Phyt}$ and $D_{Phyt}$ were also observed in Lake Chenghai. The analyses for summer (Figures S6 and S7) revealed similar results as the annual data. No clear differences were observed among the three taxa as to changes in $B_{Phyt}$ and $D_{Phyt}$ with TN, TP, and Cond between the summer and annual data (Figures 3 and S3d–l).

Percentage data on density and biomass were analyzed for the two dominant phyla by numbers: cyanophyta ($D_{Cyan}$%, $B_{Cyan}$%) and chlorophyta ($D_{Chlo}$%, $B_{Chlo}$%) (% of the sum of the two phyla) (Figure 4). $D_{Cyan}$% first increased and then leveled off with increasing TN, TP, and Cond. The point of shift from chlorophyta to cyanophyta dominance along the gradients of TN and Cond was identified as ca. 0.3 mg/L for TN (Figure 4a) and ca. 300 μS/cm for Cond (Figure 4c). Similar shifts were observed for percentage biomass and TP (Figure 4d–f). The shifting points occurred at approximately 0.6 mg/L for TN (Figure 4d), 0.07 mg/L for TP (Figure 4e), and 460 μS/cm for Cond (Figure 4f). Similar patterns were found for summer, but no clear intersection point appeared for density, reflecting on overall dominance of cyanophyta (Figure S8a–c). However, the shifting point to cyanophyta dominance of biomass was found to be similar to the annual results for TN and TP, but it occurred at approximately 300 μS/cm for Cond (Figure S8f).

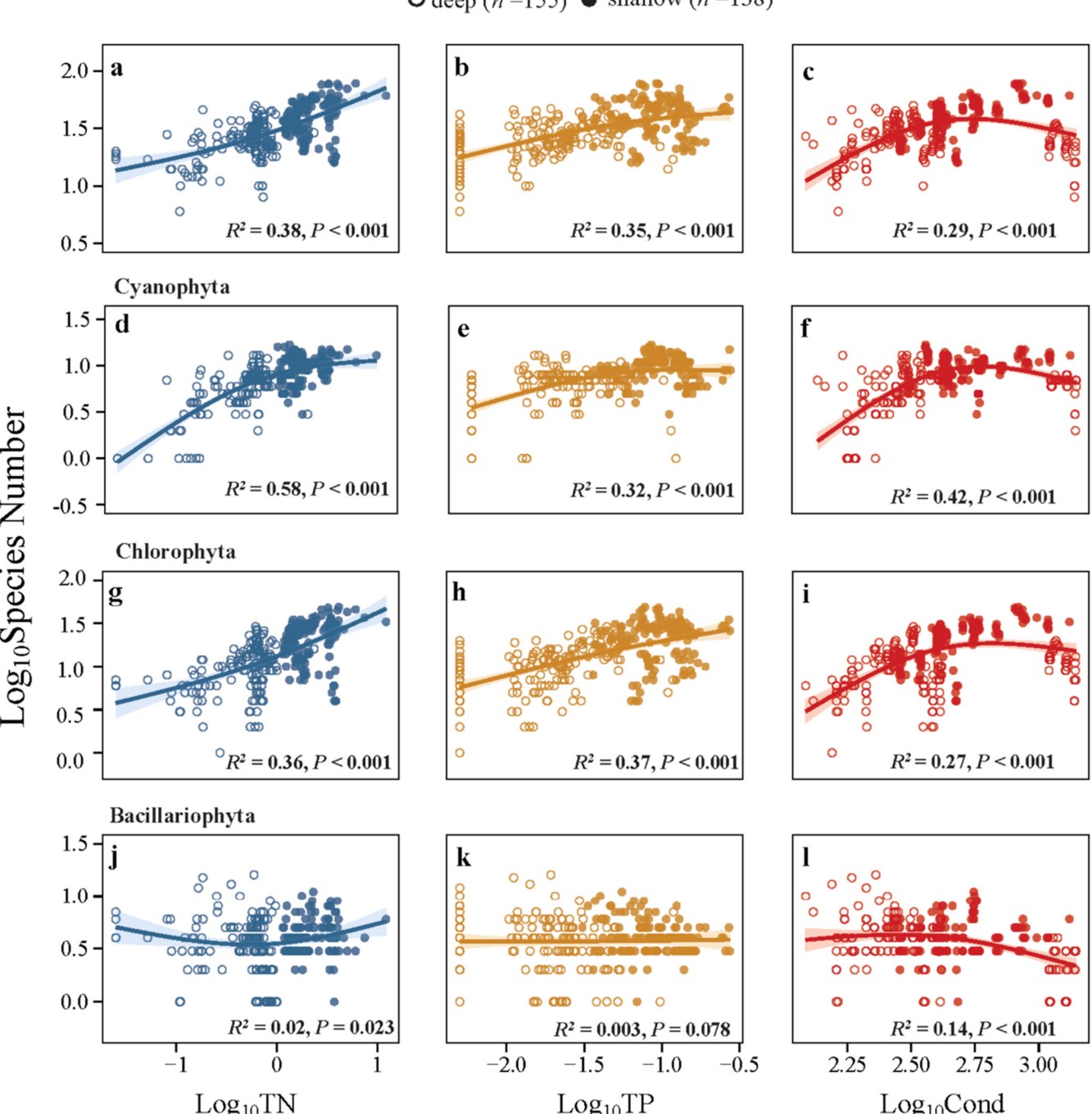

**Figure 2.** Generalized additive model showing the relationships between TN, TP, and Cond and species number of total phytoplankton (**a–c**), cyanophyta (**d–f**), chlorophyta (**g–i**), and bacillariophyta (**j–l**) in the studied lakes. The studied lakes are divided into two groups according to mean depth (Z): DC, XYH, YLH, and QLH = shallow lakes with Z < 10 m, LGH, CH, YZH, and EH = deep lakes with Z > 10 m. For model parameters see Table S1.

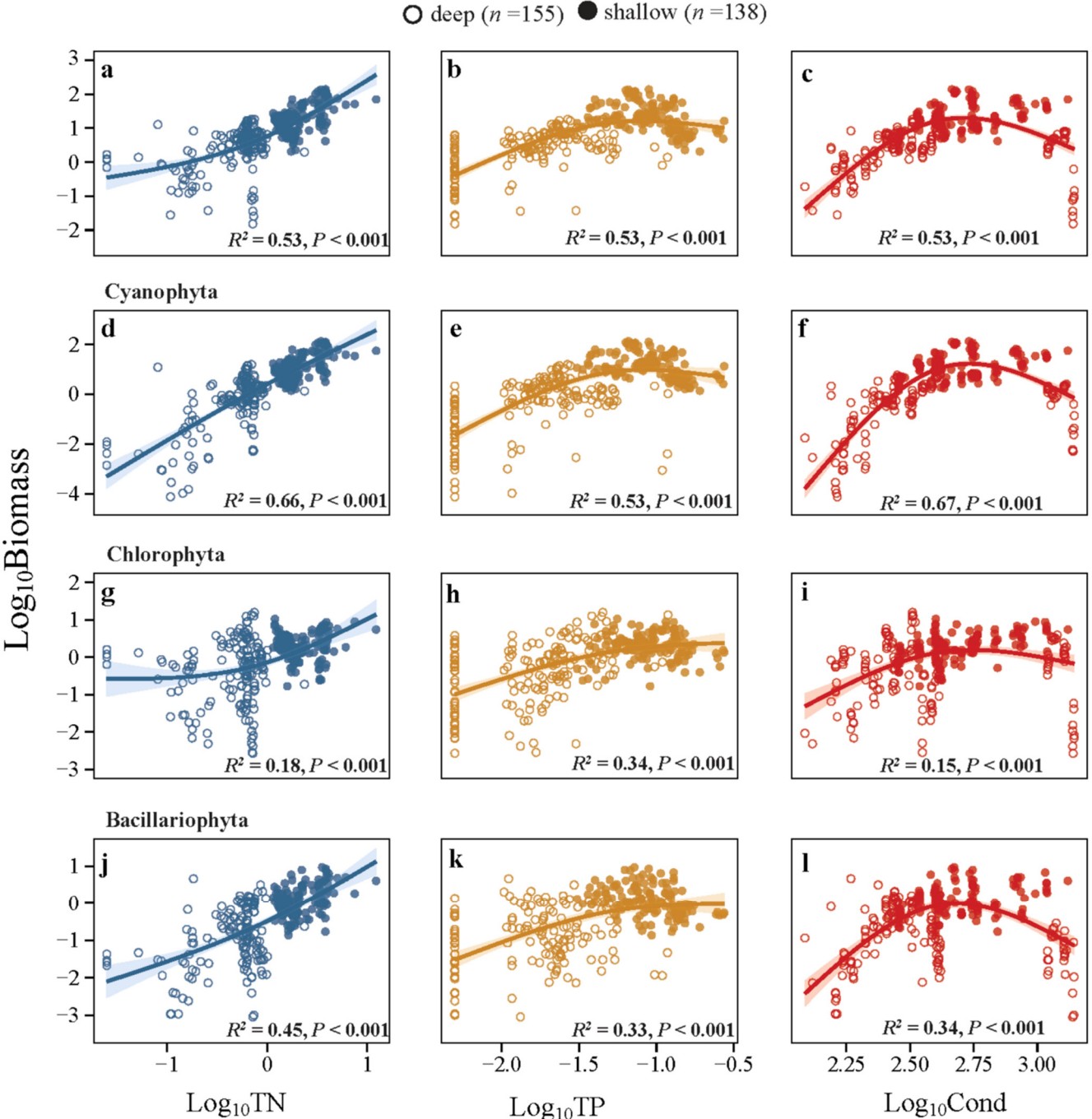

**Figure 3.** Generalized additive model showing the relationships between TN, TP, and Cond and total biomass (**a**–**c**), cyanophyta (**d**–**f**), chlorophyta (**g**–**i**), and bacillariophyta (**j**–**l**) in the studied lakes. See Figure 3 and Table S1 for model parameters used in the classification of deep and shallow lakes.

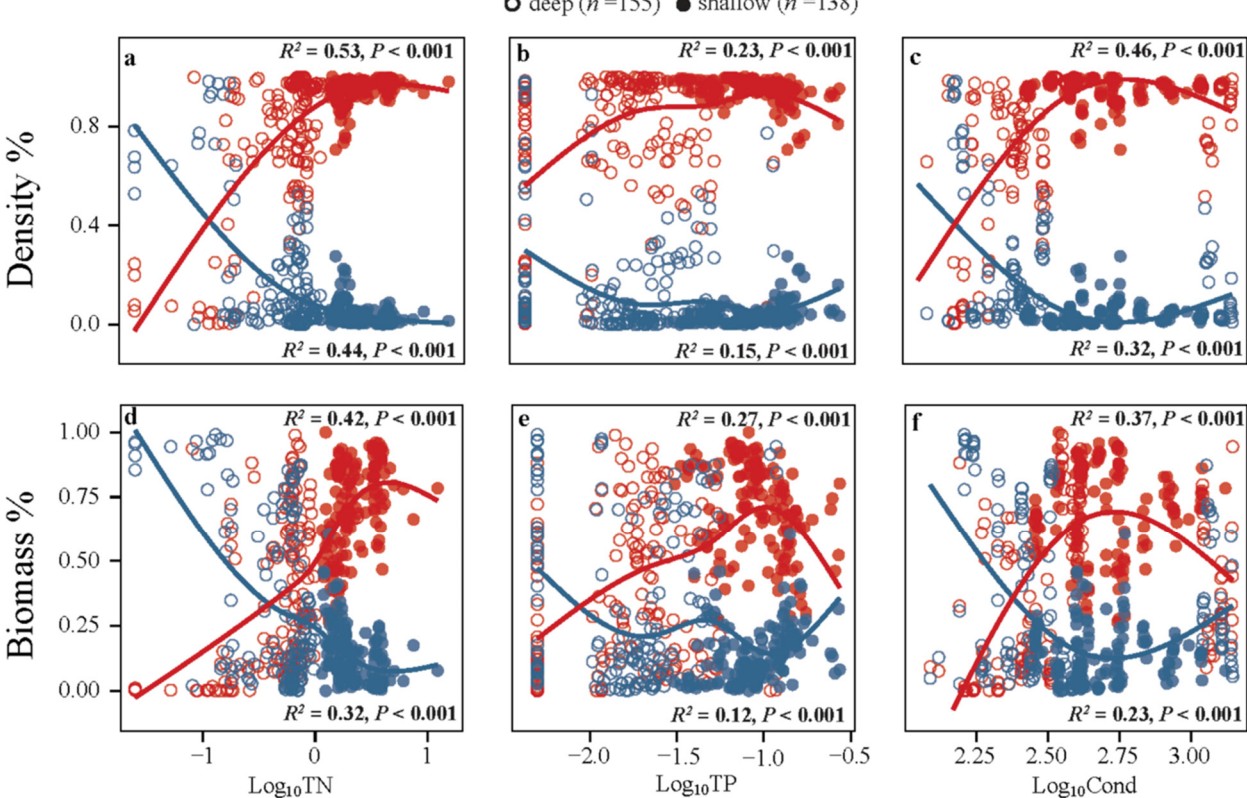

**Figure 4.** Generalized additive model measurements of the percentage share relationships of TN, TP, and Cond with density (**a**–**c**) and biomass (**d**–**f**) of cyanophyta (red) and chlorophyta (blue) in the studied lakes. See Figure 3 and Table S1 for model parameters used in the classification of deep and shallow lakes.

Four environmental variables (TN, TP, $Z_M$, and Cond) and two phytoplankton variables (total species number and biomass) were selected for multiple stepwise regression analysis (Table 2). According to the results, the final model affecting species number contained three variables, $Z_M$, TN, and Cond ($p < 0.05$). The final model for biomass contained three variables, TN, TP, and $Z_M$ ($p < 0.001$).

**Table 2.** Results of multiple stepwise regression analysis of phytoplankton and environmental parameters in the eight studied lakes.

| | Non-standardized coefficient | | Standardization coefficient | $t$ | $p$ | VIF | $R^2$ | adjusted $R^2$ | $F$ | D-W |
|---|---|---|---|---|---|---|---|---|---|---|
| | B | SE | Beta | | | | | | | |
| $Z_M$ | −0.196 | 0.023 | −0.451 | 8.544 | <0.001 | 1.54 | | | | |
| TP | 0.117 | 0.023 | 0.283 | 5.138 | <0.001 | 1.677 | 0.482 | 0.477 | 88.838 ($p < 0.001$) | 0.689 |
| Cond | 0.083 | 0.035 | 0.106 | 2.358 | 0.019 | 1.124 | | | | |
| dependent variable: Species Number | | | | | | | | | | |
| TN | 0.591 | 0.112 | 0.363 | 5.263 | <0.001 | 3.143 | | | | |
| TP | 0.42 | 0.09 | 0.273 | 4.65 | <0.001 | 2.27 | 0.566 | 0.562 | 124.511 ($p < 0.001$) | 0.674 |
| $Z_M$ | −0.326 | 0.093 | −0.202 | 3.528 | <0.001 | 2.155 | | | | |
| dependent variable: Biomass | | | | | | | | | | |

*Multiple Stepwise Regression Analysis ($n = 293$)*

## 4. Discussion

### 4.1. Changes in Species Number and Abundance of Phytoplankton

In addition to the fact that the four phytoplankton variables (SN, $D_{Phyt}$, $B_{Phyt}$, and Chl $a$) generally displayed an increasing trend with nutrient content (TN and TP) in the eight lakes of the Yunnan Plateau, agreeing with widely reported results for other lakes [20,44–46], we found a leveling of the slope at high TP, indicating the presence of other limiting factor(s) at high TP. Here, relative shortage of nitrogen was a good candidate since eutrophic lakes generally have lower N:P ratios than non-eutrophic lakes [47,48]. Accordingly, we found a gradual decline in the N:P ratio with increasing TP (see Figure S3 for the relationship between N:P ratio with TP, $Z_M$, and Cond), with the lowest ratio being 13 in Lake Xingyunhu with the highest TP (0.16 mg/L). A study of 137 lakes located within the state of Iowa, USA showed that N limitation occurred when N:P (TN: TP) by mass was lower than 20 [49]. Another study of 106 lakes from oligotrophic, boreal, sub-alpine lakes showed similar results (TN:TP < 19) [50].

We further found water depth to be an important influencing factor. $Z_M$ was significantly and negatively correlated with the number of phytoplankton species and biomass. The shallowest lake in our study was in a eutrophic state, while the deepest lake had lower nutrient levels (mean depth: 2.2–38.6 m; TN: 0.19–3.99 mg/L; TP: 0.01–0.16 mg/L). Others have also highlighted the role of lake depth. An analysis of data from 1151 lakes in America and Europe showed that lake depth amplified the sensitivity of shallow lakes to anthropogenic disturbance (mean depth: 0.5–350 m; TN: 0.014–26.1 mg/L; TP: 0.001–4.68 mg/L) [51]. A study of 30 lakes in the Yangtze River showed that the TP thresholds for a regime shift between a clear-water state dominated by submersed macrophytes and a turbid-water state dominated by phytoplankton did not vary much at moderate depths, but decreased significantly when the depth exceeded 3–4 m and increased sharply when the depth was below 1–2 m [52]. Results of a piecewise model based on 365 German lakes showed that nitrogen limitation was more common in shallow lakes than in deep lakes (mean depth: 0.0–29 m; TN: 0.36–9.18 mg/L; TP: 0.01–1.11 mg/L) [53].

Along the Cond gradient, both species number and phytoplankton abundance showed a declining trend when Cond was ca. 400-1000 μS/cm (equivalent to 91–276 mg Cl/L) in the shallow lakes, while they declined rapidly at a similar Cond level (1100 μS/cm or 336 mg Cl/L) in deep lakes. Furthermore, our results suggested that Cond was a key factor influencing the number of phytoplankton species, while the Cond effect was relative weak on biomass. The results of long-term data analysis in Fujian Province, China (0–12,000 μS/cm) showed that increased salinity decreased plankton diversity [54]. In addition, an outdoor experiment conducted at the Rensselaer Aquatic Laboratory (Troy, NY) and mesocosm experiments in high-elevation lakes in California showed that phytoplankton responded to Cond changes primarily indirectly through changes in the top-down effects of zooplankton grazing [55,56]. The high salinity tolerance of cyanobacteria [22] and the dominance of cyanobacteria in our study lakes may explain the modest effect of salinity on phytoplankton abundance/biomass in these lakes.

In natural lake ecosystem, factors normally interact with each other. Mesocosm experiments showed that phytoplankton richness increased with chloride levels at low nutrient levels and decreased with chloride levels at high nutrient levels [57]. Another mesocosm experiment suggested that the potential impacts of salinization can, to some extent, be buffered in eutrophic ecosystems and that communities developing under eutrophic conditions may be less sensitive to salinization due to cross-tolerance effects [58]. For example, results of field investigations in 20 high-elevation lakes in Qinghai (TN: 0.85–4.20 mg/L; TP: 0.10–0.21 mg/L; salinity: 0.10–200.00‰) showed that phytoplankton abundance and biomass were lower in high-salinity lakes, but high nutrient levels (especially N) helped alleviate the negative effect of salinity to some extent [59]. Additionally, the results of field investigations of 20 lakes in Siberia (depth: 1.10–24.20 m; TP: 0.01–1.58 mg/L; salinity: 0.00–40.00 g/L) showed that the interaction between eutrophication and salinization depended to a large extent on the size and depth of the lake, and that for deep lakes, salinity

enhanced stratification stability and mitigated the consequences of eutrophication [28]. The buffering effects of eutrophication may, therefore, perhaps explain the lower phytoplankton species number and phytoplankton abundance in deep lakes compared to those in shallow lakes at similar high Cond levels recorded in our study, since the deep lakes were in oligotrophic or mesotrophic states, while the shallow lakes were eutrophic.

*4.2. Changes in the Dominance of the First Two Main Phytoplankton Taxa*

We found that the dominant phytoplankton taxa by numbers, cyanophyta and chlorophyta, showed clearly different trends along the TN and Cond gradients, with chlorophyta being dominant at the lower levels and cyanophyta being dominant at the higher levels of the two environmental variables. Such shifts in dominance between cyanophyta and chlorophyta along a Cond gradient, at ca. 1275 μS/cm (400 mg Cl/L) [60], have also been reported in mesocosm experiments. Moreover, a study on the Kenyan Rift Valley showed the dominance of chlorophyta in Lake Oloidien with low Cond (280–370 μS/cm), while cyanophyta dominated in Lake Naivasha with high Cond (3890–5270 μS/cm) [61]. A study of Siberian lakes also revealed the higher dominance of cyanophyta in lakes with Cond > 1000 μS/cm than in those with Cond < 1000 μS/cm [28]. Differences in optimal growing conditions and tolerance to salt stress, as well as variations in the ion composition of the salt, may explain these contrasting responses [62,63]. Cyanophyta are more tolerant than chlorophyta to salinity stress, as suggested by various mesocosm experiments [21,22,60]. A shift in taxa dominance was also found for biomass in summer but not in terms of density where cyanophyta generally dominated along the entire environmental gradient, concurring with the fact that cyanobacteria are favored at high temperatures [4,13,64].

**5. Conclusions**

1.  Species number and abundance of phytoplankton increased with increasing TN and TP concentrations; however, the increasing trends declined at high TP concentrations in shallow lakes. Shortage of nitrogen might be the limiting factor as suggested by a low TN/TP ratio in the high-TP lakes.
2.  Phytoplankton species richness and abundance did not decline in shallow lakes but did so in deep lakes at similar Cond levels. This could perhaps be explained by the buffering effects of eutrophication since the deep lakes were in oligotrophic or mesotrophic states, while the shallow lakes were eutrophic.
3.  The two dominant phytoplankton taxa by numbers, cyanophyta (Cyan) and chlorophyta (Chlo), showed clearly different trends along the nutrient and Cond gradients, with Chlo being dominant at the lower end and Cyan being dominant at the higher ends of the environmental gradients. In summer, similar shifts in taxa dominance occurred based on biomass data but not for density, where Cyan dominated along the entire nutrient and Cond gradients.

**Supplementary Materials:** The following supporting information can be downloaded at: https: //www.mdpi.com/article/10.3390/w15010069/s1. Tables S1–S3: Results of GAM model parameters; Figure S1: Phytoplankton assemblage structure in the eight studied lakes; Figure S2: Spearman correlation analysis of the relationship between phytoplankton and environmental factors; Figure S3: Generalized additive model results showing the relationships between TN, TP, and Cond and density; Figure S4: Generalized additive model results showing the relationships between TP, $Z_M$, and Cond and TN/TP; Figures S5–S8: Generalized additive model showing the summer relationships between TN, TP, and Cond and phytoplankton variables; Figures S9–S12: Generalized additive model of the relationships between TN, TP, and Cond and phytoplankton variables based on lake-specific data of four seasons.

**Author Contributions:** Conceptualization, Y.W., H.-J.W., Y.L. and P.X.; methodology, Y.W., L.-J.Y., Y.-H.L., L.Z., P.-Z.W. and X.Z.; formal analysis, Y.W., L.-J.Y.; writing—original draft preparation, Y.W.; writing—review and editing, H.-J.W., Y.-L.L., X.J., P.X. and E.J.; supervision, H.-J.W. and E.J. All authors have read and agreed to the published version of the manuscript.

**Funding:** This research was supported by the Yunnan Provincial Department of Science and Technology (202103AC100001; 202001BB050078) and the Strategic Priority Research Program of the Chinese Academy of Sciences (XDB31000000). EJ was supported by the TÜBITAK program BIDEB2232 (project 118C250).

**Institutional Review Board Statement:** Not applicable.

**Informed Consent Statement:** Not applicable.

**Data Availability Statement:** The data presented in this study are available in Supplementary Materials.

**Acknowledgments:** The authors would like to thank Hui-Lin Bi, Yuan-Yuan Li for helping with the experiments, Ye-Xin Yu, Qing-Yang Rao, Lei Shi, Yan-Feng Sun, and Hao-Jie Su for their help in data analysis. We thank Anne Mette Poulsen for valuable English editions.

**Conflicts of Interest:** The authors declare no conflict of interest.

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
