# Peer review of "Interactive Effects of Nutrients and Salinity on Phytoplankton in Subtropical Plateau Lakes of Contrasting Water Depths"

_water, doi:10.3390/w15010069_

Round 1

Reviewer 1 Report

Overall, this manuscript is very well written and focuses on important matters regarding nutrients and salinity effects on phytoplankton communities. I believe that the manuscript is suitable for acceptance, however, small changes are required, and more importantly, further discussion of results is mandatory.

Introduction

Overall well written, and concise.

Line 50: “Salinization also favors the growth of cyanobacteria”, how? Should be further described in this section.

Material and Methods

Line 101-103: Do you have references for these? If not, please detail better these methodologies, and indicate which spectrophotometer.

Line 119: remove comma “[42].,”

Results

Line 148: “phytoplankton generally increased with nutrient status (Fig-148 ure 2 c, d)” I cannot see the nutrient status in the figure, the way that it is written in the text gives the idea that you can see the synergy between phytoplankton and nutrient. I would suggest altering the text, the figure is ok.

Line 156 – Figure2: The caption can be quite confusing, sometimes figure sections come after the explanation, sometimes before… I would suggest: “Phytoplankton assemblage structure in the eight studied lakes. Species number of 157 the eight major taxa (a) and species number in the four seasons (b). Density (c) and biomass (d) of the 158 main phytoplankton groups, and percentage shares of density (e) and biomass (f) of phytoplankton in 159 the studied lakes in the four seasons…”. Suggestion, 1a – enlarge the Eight phyla text in the graph.

Line 168: “(see appendix), 168 showing overall showed similar results as the total dataset, see annexes Figures S8-S11.” Remove the “(see appendix)” it is already written to see the figures S8-S11.

Line 194: “The studied lakes re divided into”, correct “are”

Line 198-199: Correct “with Biomass 198 (a-c), total (a-c),” to “with total biomass (a-c)”

Line 208: “460 μS/cm for Cond (Figure 5 c).” do you mean Figure 5 f?

Discussion

Overall could be improved, the results could be further discussed and compared with many other already published related works. As is referred to in the introduction, it would be expected to see more discussion around salinity. Some more discussion around lake depth comparisons should also be done, which are the focus of this work.

References

Since further discussion of results is encouraged, the references list should also be enhanced.

Author Response

General comments:

Comment 1: Small changes are required, and more importantly, further discussion of results is mandatory.

Response: Thank you for your suggestion, we have double-checked the manuscript, added new results and discussions, and improved the entire text.

Introduction

Comment 2: Line 50: “Salinization also favors the growth of cyanobacteria”, how? Should be further described in this section.

Response: We have made the appropriate modifications, please see line 50-53.

Materials and methods

Comment 3: Line 101-103: Do you have references for these? If not, please detail better these methodologies, and indicate which spectrophotometer.

Response: Yes, we do. The reference is Water and Wastewater Monitoring and Analysis Methods of the Ministry of Environmental Protection of the People’s Republic of China, which was cited as number 35. The methods and spectrophotometers were also supplemented with appropriate, please see line 101-106.

Comment 4: Line 119: remove comma “[42].,”

Response: Done as suggested.

Results

Comment 5: Line 148: “phytoplankton generally increased with nutrient status (Figure 2 c, d)” I cannot see the nutrient status in the figure, the way that it is written in the text gives the idea that you can see the synergy between phytoplankton and nutrient. I would suggest altering the text, the figure is ok.

Response: The individual lakes in the figure are sorted from left to right according to nutrient status from lowest to highest, so the variation with nutrient status is mentioned in the description. These descriptions were included in the revision to make it easier to understand. Please note that, based on another reviewer's suggestion, we put Figure 2 as the basic supplementary information in the annex in order to make the graphical information of the article more concise.

Comment 6: Line 156 – Figure2: The caption can be quite confusing, sometimes figure sections come after the explanation, sometimes before… I would suggest: “Phytoplankton assemblage structure in the eight studied lakes. Species number of 157 the eight major taxa (a) and species number in the four seasons (b). Density (c) and biomass (d) of the 158 main phytoplankton groups, and percentage shares of density (e) and biomass (f) of phytoplankton in 159 the studied lakes in the four seasons…”. Suggestion, 1a – enlarge the Eight phyla text in the graph.

Response: Done as suggested.

Comment 7: Line 168: “(see appendix), 168 showing overall showed similar results as the total dataset, see annexes Figures S8-S11.” Remove the “(see appendix)” it is already written to see the figures S8-S11

Response: Done as suggested.

Comment 8: Line 194: “The studied lakes re divided into”, correct “are”

Response: Done as suggested.

Comment 9: Line 198-199: Correct “with Biomass 198 (a-c), total (a-c),” to “with total biomass (a-c)”

Response: Done as suggested.

Comment 10: Line 208: “460 μS /cm for Cond (Figure 5 c).” do you mean Figure 5 f?

Response: Yes, we did mean Figure 5f. It’s replaced in the revision.

Discussion

Comment 11: Overall could be improved, the results could be further discussed and compared with many other already published related works. As is referred to in the introduction, it would be expected to see more discussion around salinity. Some more discussion around lake depth comparisons should also be done, which are the focus of this work.

Response: We agree with your comments. In the section of Results, we have added further analyses of multiple stepwise regressions, which showed the relative contributions of nutrients, water depth, and conductivity to explaining the variations of phytoplankton among lakes. In the section of Discussions, some more were also included, particularly on the role of salinity and water depth, as suggested (please see lines 246-255, 281-327).

References

Comment 12: :Since further discussion of results is encouraged, the references list should also be enhanced.

Response: Done as suggested.

More details , please see the attachment.

Reviewer 2 Report

Understanding how human activity can impact phytoplankton communities in lakes is of great interest given rapidly changing environmental conditions.  While the introduction lays out sufficient reason for why a study like this needs to be undertaken, the results and discussion fail to present the results in a coherent manner. 

For example, Figure 2 is very difficult to understand.  The panels, multiple colors, and small writing makes it very difficult to interpret exactly what the authors would like the reader to notice.  I think the data is there, it just needs to be presented better.  Can things be divided into seasons instead of putting all seasons on one graph?  Can some lake sites be grouped together for visualization to compare oligotrophic v mesotrophic v eutrophic?  The graphics are simply too cluttered to be understood. 

The introduction laid out an argument for understanding the synergistic impacts of eutrophication and salinization, but that is lost in the reporting.  If that is the true focus of the paper, can the figures be reconfigured to demonstrate that point of view more clearly?  

The main problem with papers that look at single samples with phytoplankton community measurements and the nutrient concentrations is the fact that you are looking at such a small snapshot in time.  The nutrients currently being sampled are not what shaped the community being measured.  It was the nutrients in the days, maybe even weeks prior that led to the current community.  

I encourage streamling of the figures being presented to really focus on what is most important to take away from this paper.  Currently, it is very difficult to understand what is trying to be presented and it is very difficult to believe the conclusions the authors are trying to come to.  

The attached file has some annotations for review.

Author Response

General comments:

Comment 1: While the introduction lays out sufficient reason for why a study like this needs to be undertaken, the results and discussion fail to present the results in a coherent manner.

Response: Thank you for your suggestion. We fully agree with your comments. In the section of Results, we have added further analyses of multiple stepwise regressions, which showed the relative contributions of nutrients, water depth, and conductivity to explaining the variations of phytoplankton among lakes. In the section of Discussions, some more were also included, particularly on the role of salinity and water depth, as suggested (please see lines 246-255, 281-327).

Comment 2: Figure 2 is very difficult to understand. The panels, multiple colors, and small writing makes it very difficult to interpret exactly what the authors would like the reader to notice. I think the data is there, it just needs to be presented better. Can things be divided into seasons instead of putting all seasons on one graph? Can some lake sites be grouped together for visualization to compare oligotrophic v mesotrophic v eutrophic? The graphics are simply too cluttered to be understood.

Response: We modified Figure 2 based on the suggestions to make it more intuitive to represent the key information. In addition, to make the article more concise, we moved figure 2 to the appendix as Figure S1.

Comment 3: The introduction laid out an argument for understanding the synergistic impacts of eutrophication and salinization, but that is lost in the reporting. If that is the true focus of the paper, can the figures be reconfigured to demonstrate that point of view more clearly?

Response: Thanks for your comments. To make the purpose and results more consistent, we have added analyses of multiple stepwise regression in the results. Furthermore, some more discussion on salinity, water depth and the synergistic impacts of eutrophication and salinization were also included in the discussion section. (please see lines 246-255, 281-327).

Comment 4: The main problem with papers that look at single samples with phytoplankton community measurements and the nutrient concentrations is the fact that you are looking at such a small snapshot in time. The nutrients currently being sampled are not what shaped the community being measured. It was the nutrients in the days, maybe even weeks prior that led to the current community.

Response: We fully agree with your comments. Actually, it is also a common problem in field investigation of limnological studies. Limited by time and cost, people normally visit the waters only once for specific field investigation. Evidence like this is truly weak in presenting the cause-effect relationships. This is particularly so when a study is performed on a specific lake. In our study, our aim was to explore the response of phytoplankton assemblages to external environmental changes based on investigating eight lakes with relative wide environmental gradients. It’s a kind of space-for-time approach, which is helpful and normally acceptable method used to understand the species-environment relationships.

Comment 5: I encourage streamling of the figures being presented to really focus on what is most important to take away from this paper. Currently, it is very difficult to understand what is trying to be presented and it is very difficult to believe the conclusions the authors are trying to come to.

Response: We have added and subtracted charts and graphs from the article in order to make the whole article clearer and hopefully to convey the results consistently and concisely.

Introduction

Comment 6: Line 39: Delete “the”.

Response: Done as suggested.

Comment 7: Line 54: Correct “for” to “on”.

Response: Done as suggested.

Materials and methods

Comment 3: Line 107: What drove the decision? Biomass in samples or were these decisions made ahead of time?

Response: It was a mistake. We feel sorry for making you confused. It was 50 mL. It was corrected in the revision.

Comment 4: Line 123: What drove this decision?

Response: The current standard water depth boundaries regarding the distinction between deep and shallow lakes are not clear. Of the eight lakes we investigated and studied, stratification existed in four lakes with an average water depth of 10 m or more, while stratification did not occur in four lakes below 10 m. Therefore, in conducting the analysis, we used 10 m as the boundary to distinguish between deep and shallow lakes. Relevant explanation was added in the revision (Please see lines 128-129).

Results

Comment 5: Table 1: Why are these two capitalized and the others lowercase? Part of the table? Can you get it on one page together?

Response: All corrected or revised as suggested.

Comment 6: Figure2: Do all season need to be on these graphs? I’m lost as to the take home message that is trying to be communicated? This is really hard to read in this format.

Response: In the paper, Figure 2 is used as a basic supplement to present the composition and abundance of phytoplankton in the lakes studied over the study period. In order to make the picture more intuitive, we’ve modified the four subgraphs a-d. To make the article more concise, Figure 2 was moved to the appendix as Figure S1.

Please refer to the attachment for details of the changes.

Round 2

Reviewer 2 Report

Thank you for following the suggestions from the different reviewers and addressing the inadequacies that were pointed out.